# bHLH transcription factors cooperate with chromatin remodelers to regulate cell fate decisions during *Arabidopsis* stomatal development

Ao Liu[1]☯, Andrea Mair[1]☯, Juliana L. Matos[2]¤, Macy Vollbrecht[2], Shou-Ling Xu[3,4], Dominique C. Bergmann [1,2]*

1 Howard Hughes Medical Institute, Stanford, California, United States of America, 2 Department of Biology, Stanford University, Stanford, California, United States of America, 3 Carnegie Institution for Science, Stanford, California, United States of America, 4 Carnegie Mass Spectrometry Facility, Carnegie Institution for Science, Stanford, California, United States of America

☯ These authors contributed equally to this work.
¤ Current address: Department of Chemical and Biomolecular Engineering, University of California, Berkeley, California, United States of America
* dbergmann@stanford.edu

**Data Availability Statement:** All data needed to evaluate the conclusions in the paper are present in

## Abstract

The development of multicellular organisms requires coordinated changes in gene expression that are often mediated by the interaction between transcription factors (TFs) and their corresponding *cis*-regulatory elements (CREs). During development and differentiation, the accessibility of CREs is dynamically modulated by the epigenome. How the epigenome, CREs, and TFs together exert control over cell fate commitment remains to be fully understood. In the *Arabidopsis* leaf epidermis, meristemoids undergo a series of stereotyped cell divisions, then switch fate to commit to stomatal differentiation. Newly created or reanalyzed scRNA-seq and ChIP-seq data confirm that stomatal development involves distinctive phases of transcriptional regulation and that differentially regulated genes are bound by the stomatal basic helix–loop–helix (bHLH) TFs. Targets of the bHLHs often reside in repressive chromatin before activation. MNase-seq evidence further suggests that the repressive state can be overcome and remodeled upon activation by specific stomatal bHLHs. We propose that chromatin remodeling is mediated through the recruitment of a set of physical interactors that we identified through proximity labeling—the ATPase-dependent chromatin remodeling SWI/SNF complex and the histone acetyltransferase HAC1. The bHLHs and chromatin remodelers localize to overlapping genomic regions in a hierarchical order. Furthermore, plants with stage-specific knockdown of the SWI/SNF components or HAC1 fail to activate specific bHLH targets and display stomatal development defects. Together, these data converge on a model for how stomatal TFs and epigenetic machinery cooperatively regulate transcription and chromatin remodeling during progressive fate specification.

the paper or in the data and databases described in S1 Data.

**Funding:** This work was supported by funds from the Cellular and Molecular Biology training grant (National Institutes of Health, T32GM007276 to Stanford University in support of MV), by the Carnegie endowment fund to the Carnegie mass spectrometry facility (S-LX) and by the Howard Hughes Medical Institute (Investigator award to DCB). The Howard Hughes Medical Institute provided salaries for AL, AM and DCB. The funders had no role in study design, data collection and analysis, decision to publish, or preparation of the manuscript.

**Competing interests:** The authors have declared that no competing interests exist.

**Abbreviations:** 3-AT, 3-amino-1,2,4-triazole; ACD, asymmetric cell division; amiRNA, artificial microRNA; ATAC-seq, assay for transposase-accessible chromatin using sequencing; AtML1, ARABIDOPSIS THALIANA MERISTEM L1 LAYER; bHLH, basic helix–loop–helix; BiFC, bimolecular fluorescence complementation; BRM, BRAHMA; ChIP-seq, chromatin immunoprecipitation followed by deep sequencing; CRE, *cis*-regulatory element; dpg, day postgermination; GC, guard cell; GMC, guard mother cell; LFQ, label-free quantification; LFY, LEAFY; PWM, position weight matrix; qRT-PCR, quantitative reverse transcription PCR; SCD, symmetric cell division; SCRM, SCREAM; scRNA-seq, single-cell RNA-sequencing; SD, synthetic dropout; SPCH, SPEECHLESS; SWI/SNF, SWITCH DEFECTIVE/SUCROSE NONFERMENTABLE; TbID, TurboID; TF, transcription factor; WT, wild type; Y2H, yeast two-hybrid.

## Introduction

Cells integrate a wide range of internal and external cues to execute complex patterns of gene expression. During development and processes such as wound healing and tissue regeneration, coordinated global changes in transcription lead to the formation of diverse types of cells with different fates. Disruption of such controlled gene expression can cause a variety of developmental and growth defects and can be lethal in extreme cases [1]. Unraveling the regulatory mechanisms of developmentally important transcriptional programs during fate specification is therefore important for understanding development and is a key step towards reprogramming cells for different applications.

With all cells possessing identical genetic information, divergence in the transcriptome during fate transition is attributed to the complex interactions between *cis*-regulatory DNA elements (CREs) and transcription factors (TFs) [2]. To initiate transcription, CREs are accessed by TFs, which then recruit cofactors to facilitate the assembly of the transcription machinery. Differential accessibility of CREs, therefore, lies at the core of executing cell type–specific gene expression programs. To regulate CRE accessibility, eukaryotes have complex and conserved mechanisms at the chromatin level. Open chromatin regions, which allow TFs to bind to the CREs, typically display low nucleosome density or nucleosome-free regions, while closed chromatin is tightly packed with a high nucleosome density [3]. Another level of regulation comes from histone variants and posttranslational modifications of histone tails within the nucleosome. Active chromatin usually displays a combination of modifications such as H3K27ac and H3K4me3, and variants of core histone proteins such as H2A.Z and H3.3 [4–7]. Transcriptionally inactive genes, on the other hand, are localized within closed chromatin regions that are usually marked by repressive histone modifications such as H3K27me3 and H3K9me3 [3,5].

Before cell fate transition, progenitor cells are primed for distinct cell identities epigenetically [6–9] and transcriptionally [10–12]. When and how such priming occurs are questions best answered by characterizing biologically relevant variations between single cells. In cell differentiation contexts, entropy has been introduced as a measure of variability, with the assumption that cell populations under strict regulatory constraints should exhibit well-defined and low entropy expression patterns, whereas those under weaker regulatory constraints should show diverse, higher entropy expression patterns [13–15]. Entropy quantification can be used as a method to identify TFs involved in priming cell fate [16].

While entropy measures can reveal state changes, we are still faced with the question how CREs in silent chromatin are first accessed and chromatin states reprogrammed. In addition to the epigenetic machinery, special "pioneer" TFs that are capable of binding nucleosomal (closed) CREs have been shown to override repressive chromatin states, initiate the remodeling of the epigenome, and pave the way for the transcriptional machinery and other TFs to activate gene expression in animal systems [17]. Such TFs are thus crucial for the initiation of new cell fates, given the compact nature of DNA packaging. In plants, such activity has so far only been reported for LEAFY (LFY) [18], a TF that is indispensable for flower development [19]. This was a landmark result, but for technical reasons, the chromatin remodeling function of LFY in fate specification was characterized in root explants, a callus-based regeneration system where sizable epigenetic reprogramming has occurred during the dedifferentiation process [20]. It remains to be shown whether plant TFs are able to reshape the native chromatin landscape in endogenous developmental pathways and, if so, how.

To address these open questions, we turned to stomatal development where a simple and stereotyped lineage exhibits clear fate transitions guided by well-characterized TFs. Stomata are leaf pores surrounded by a pair of epidermal guard cells (GCs). Fate transitions during stomatal development require the activity of heterodimeric basic helix–loop–helix (bHLH) TF

complexes consisting of one of 3 stage-specific bHLHs—SPEECHLESS (SPCH), MUTE or FAMA, and either SCREAM (SCRM, also ICE1) or SCRM2, 2 redundant partners expressed throughout the lineage and required at each stage [21]. Expression of SPCH marks the entry into the stomatal lineage, creating a meristemoid that is capable of self-renewing through asymmetric cell divisions (ACDs) [22,23]. Activation of MUTE puts an end to ACDs and the transition from meristemoids to guard mother cell (GMC) fate [23]. The GMC undergoes a single symmetric cell division (SCD) and, under the guidance of FAMA, creates a pair of cells that differentiate into the GC complex [24]. Misregulation of each stage-specific stomatal bHLH factor has a significant and characteristic impact on cell fate and development. Overexpression of SPCH induces precursor cell overproliferation, whereas MUTE overexpression forces epidermal cells to adopt GMC-like fates and subsequently divide into paired stomatal guard cells [22,23]. Similarly, the gain-of-function *scrm-D* allele causes all epidermal cells to become guard cells [21]. Guard cell differentiation remains dependent on intact *FAMA* activity in these situations, and overexpression of FAMA can also promote transdifferentiation into guard cell identity, albeit without cell division, so the stomata are not functional [24]. The competence of MUTE, FAMA, and SCRM to direct cells toward guard cell fate makes them appealing candidates in breaking the barrier of repressive chromatin to activate gene expression during stomatal development.

By analyzing transcriptional entropy in stomatal lineage single-cell RNA-sequencing (scRNA-seq) data, we identify different phases of transcriptional regulation during stomatal development. Furthermore, we show that MUTE, FAMA, and SCRM indeed can overcome repressive chromatin, and each has the potential to induce nucleosome repositioning during these phases. Chromatin remodeling is potentially mediated by the SWITCH DEFECTIVE/ SUCROSE NONFERMENTABLE (SWI/SNF) complex and the histone acetyltransferase HAC1, both of which physically associate with the bHLHs in stomatal lineage cells. Stomatal lineage-specific knockdown of components of the SWI/SNF complex or HAC1 results in reduced expression of the bHLHs' targets and in plants that are impaired in forming terminally differentiated stomata. Together, these studies provide compelling evidence of cell type–specific chromatin remodeling during plant development and lay the foundation for studying the mechanism of how dynamic interactions between CREs, TFs, and epigenetic machinery mediate fate transitions in plants.

## Results

### Stomatal development involves different waves of transcriptional regulation

Recent advances in single-cell techniques allow the construction of developmental trajectories and have revealed a high degree of cellular heterogeneity and rapid switches in cellular state transitions. Although stochasticity and variable population sizes still remain challenges, a substantial part of the expression variation has been shown to be functionally important [25]. Such biologically relevant heterogeneity can generally be attributed to cells in states of different plasticity [13–15]. Cellular plasticity can be measured using transcriptional entropy, which quantifies the degree of uncertainty of the expression levels of all genes in a cell, and can provide a robust proxy for the differentiation potential of a cell [13]. Transcriptional entropy is calculated as the sum of the expression probabilities of each gene in the cell. In pluripotent animal cells with weak transcriptional regulation, more genes are expressed in a relatively homogeneous way, resulting in high entropy, while in more differentiated cells, transcriptional regulation is more constrained, displaying well-defined and low entropy expression patterns (illustrated in Fig 1A). To evaluate whether a similar situation exists in differentiating plant

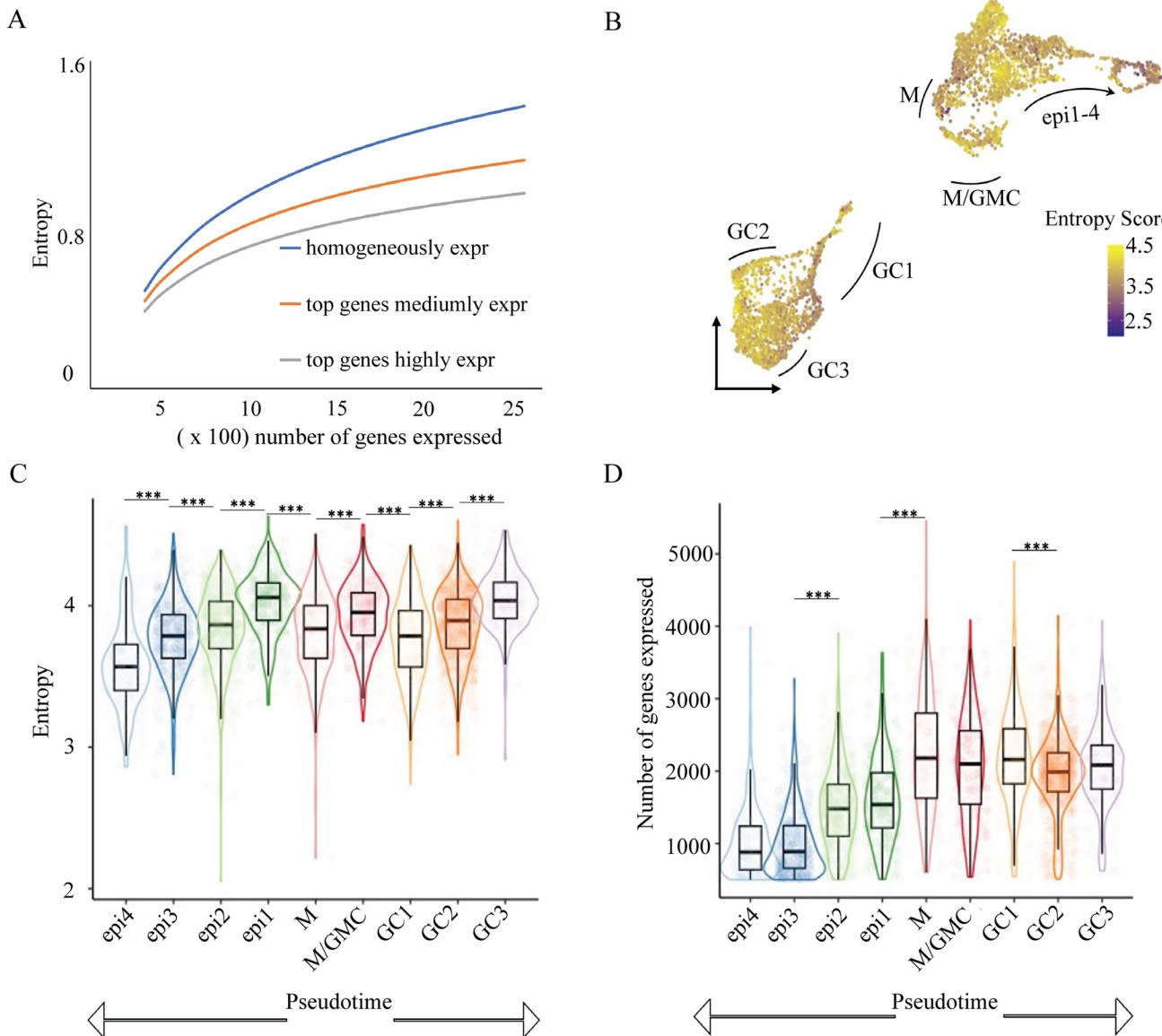

**Fig 1. Dynamic changes in transcriptional entropy and number of genes expressed during stomatal development.** (**A**) The principle of transcriptional entropy is illustrated by comparing the correlation of entropy and expressed genes for simulated cell populations with homogenous gene expression (same expression for all genes, blue line) and populations in which the top 10% of expressed genes comprise 40% (orange line) or 50% (grey line) of all transcripts. An increase in the number of expressed genes leads to a rise in entropy, whereas increased expression heterogeneity results in a drop in entropy. (**B**) Heatmap of entropy scores overlaid on a UMAP plot of scRNAseq data from epidermal cells of developing leaves. Data from [27]. (**C, D**) Boxplot of entropy scores (**C**) and number of genes expressed (**D**) in different epidermal cell types. Cell types are indicated in panel keys with M (meristemoids) being the most pluripotent cells and arrows below the plots indicating bidirectional paths to differentiation as guard cell (GC) or epidermal pavement cells (epi). The data underlying this figure can be found in S1 Data.

tissues, we first analyzed scRNA-seq data from roots [26], as roots display clear unidirectional differentiation trajectories, and found a general trend of decreased entropy along trajectories of multiple root cell types (S1A Fig). Most root cell types exhibit the highest entropy in the meristematic zone (S1B Fig).

We then asked whether a similar pattern of entropy decreasing with differentiation emerges in the stomatal lineage [27]. We found dynamic changes in transcriptional entropy, as well as

the number of genes expressed in different leaf epidermal cell types (Fig 1B–1D). In general, entropy decreases from the most pluripotent (meristemoid; M) stages toward the most differentiated stages (epidermis; epi and guard cell; GC), although interestingly, the fate commitment of GCs involves more than one step of transcriptional regulation. Upon entering GC fate (GC1), cells display a significant decrease in transcriptional entropy, while the total number of genes expressed does not significantly drop (Fig 1C and 1D), suggesting a dramatic up-regulation of a specific subset of genes that greatly skews the expression profile, resulting in a low entropy score. The transcriptional entropy then increases along the developmental trajectory of GCs (GC2 and GC3). A rise in entropy can either be caused by an increase in the number of expressed genes or by a switch to a more uniform expression pattern (Fig 1A). As the number of expressed genes decreases during the progression from GC1 through GC3 (Fig 1D), the increased entropy score indicates that the expression level of those genes is becoming more homogenous.

## Regulation of differentially expressed genes during fate transition is a battle against closed chromatin

The drop in the entropy score upon the transition to GC fate led us to explore which factors contribute to the change in transcript complexity at this stage. Given the importance of MUTE, FAMA, and SCRM for commitment to GC fate, we reasoned that they may play a critical part in mediating gene expression dynamics. We first characterized in vivo binding patterns of the stomatal bHLHs by analyzing published chromatin immunoprecipitation followed by deep sequencing (ChIP-seq) data. This included published data for SPCH, MUTE, and SCRM [28–30] and a newly generated ChIP-seq dataset of FAMA (see Materials and methods). For each data set, peaks and peak summits were defined by MACS2 [31]. We compared the peaks statistically by random permutation using ChIPseeker [32] and found a significant overlap of binding sites among these bHLHs (S2A Fig) as well as a shared G-box binding motif within 100 bp of peak summits (S2B Fig). These results suggest conserved DNA binding specificity and potentially shared functions between these 4 TFs. We then compared the genes that are associated with the bHLH peaks with those that are differentially expressed at individual stages of stomatal development as determined by scRNA-seq. We found a large proportion of genes that were differentially expressed during fate transitions were bound by a bHLH in ChIP-seq (S1 Data).

To further explore transcriptional regulation during the fate transitions, we characterized the epigenetic states of bHLH binding sites with differential expression patterns by examining their chromatin accessibility. We first queried published tissue-specific assay for transposase-accessible chromatin using sequencing (ATAC-seq) datasets of tissues derived from meristemoids [33] and from differentiated tissues [34]. We found that MUTE-, FAMA-, and SCRM-targeted CREs are less accessible than SPCH targets or random sites in bulk tissues (Figs 2A and S3B). We then characterized the chromatin state dynamics of identified bHLH targets that are differentially expressed in the context of stomatal development. For this, we compared histone modifications of these loci in mature GCs and in reprogrammed GCs that had been reverted to an early, SPCH-expressing, state via disruption of RBR interaction ($FAMA^{LGK}$, published in [35–37]). In reprogrammed GCs, targets of MUTE, FAMA, and SCRM, but not of SPCH, were highly H3K27-trimethylated (Fig 2B–2D). This suggests that MUTE and FAMA, together with SCRM, access and regulate genes that are buried in silenced chromatin during early stomatal lineage stages. In wild-type GCs, H3K27me3 decreased specifically at MUTE, FAMA, and SCRM targets, but not at SPCH targets or randomly sampled genes (Fig 2D). These dynamics suggest that chromatin at MUTE, FAMA, and SCRM targets undergoes

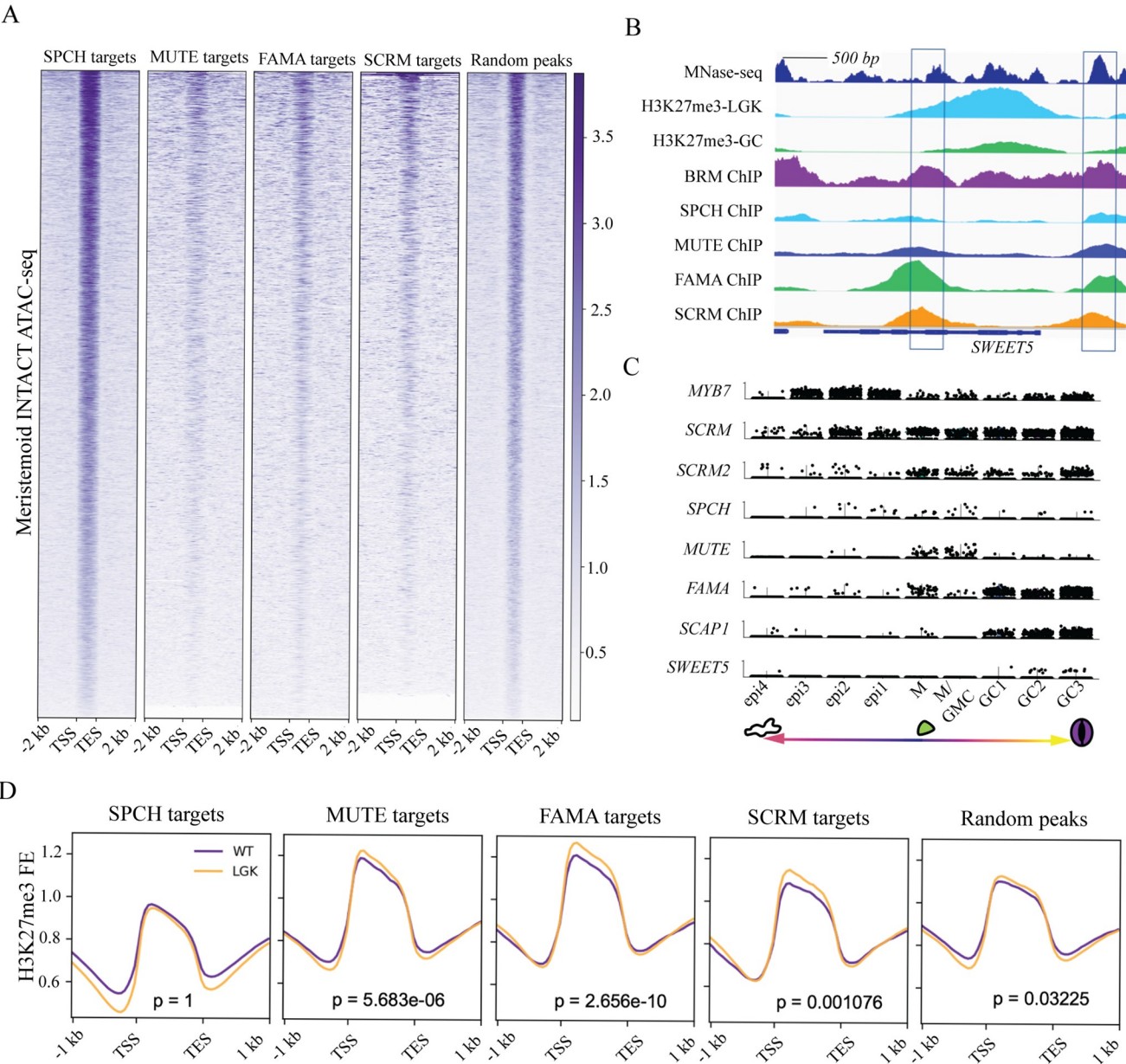

**Fig 2. Differentially regulated genes reside in closed chromatin regions and undergo dynamic changes in chromatin signature during fate transitions. (A)** Openness of the chromatin determined by ATAC-seq [33] read enrichment at SPCH [28], MUTE [29], FAMA, and SCRM [30] targets and random intergenic intervals in meristemoids. Open regions are dark purple; closed regions are white. **(B)** Example of a shared bHLH target (SWEET5) that is nucleosome bound (MNase-seq [38]; top track, deep blue), heavily H3K27me3-methylated in early stomatal lineage cells (H3K27me3 ChIP-seq in FAMA[LGK] cells [37]; second track, light blue) and shows a decrease in H3K27me3 in mature GCs (H3K27me3 ChIP-seq in GC [37]; third track, green); the bottom 5 tracks show binding of the stomatal lineage bHLHs and the chromatin remodeler BRM from ChIP-seq experiments. **(C)** Late stomatal lineage-specific expression pattern of SWEET5 in scRNA-seq. Violin plots show the expression pattern (cells with expression) of cell type markers (top 7) and SWEET5 (bottom) in the following cell clusters: alternative epidermal (pavement cells) fate (epi1-4), meristemoid (M and M/GMC), guard cell (GC1-3). **(D)** Genome-wide average fold enrichment of H3K27me3 at the gene body in WT GC (purple) and in FAMA[LGK] cells (reprogrammed GCs with earlier lineage identity, orange) at SPCH, MUTE, FAMA, and SCRM targets and random genes. The data underlying this figure can be found in S1 Data. ATAC-seq, assay for transposase-accessible chromatin using sequencing; bHLH, basic helix–loop–helix; ChIP-seq, chromatin immunoprecipitation followed by deep sequencing; GC, guard cell; SCRM, SCREAM; scRNA-seq, single-cell RNA-sequencing; SPCH, SPEECHLESS; TES, target end sites; TSS, target start sites; WT, wild type.

a transition from a repressive state to an active state as they progress from precursor to committed and mature stomatal fates (Fig 2B–2D). Levels of the active mark H3K4me3 did not change at the same targets in the wild-type GCs (S3A Fig), consistent with MUTE, FAMA, and SCRM targets being in a generally repressed state. We reasoned that to activate these genes during cell fate transitions, MUTE, FAMA, and SCRM must access repressive chromatin regions and eventually promote a reduction in chromatin compactness as indicated by histone modifications.

## bHLHs have the potential to bind closed chromatin regions and can induce chromatin remodeling

To determine whether MUTE, FAMA, and SCRM have the potential to bind to closed chromatin conformations, we analyzed nucleosome density at their respective binding sites using published MNase-seq datasets [38]. To put these data in context, we used LFY, a known nucleosome binder as a positive control, and PIF4, a bHLH factor not known to bind nucleosomes, as a negative control. Nucleosome density was comparable among LFY, FAMA, and SPCH targets, but MUTE and SCRM targets displayed a much higher nucleosomal signal (Fig 3A). Nucleosome density at targets of PIF4 [39] was negligible (Fig 3A). These data suggest that high nucleosome density is not a general feature of the target sites of *Arabidopsis* bHLH TFs but may be a specific feature of the stomatal lineage TFs.

Nucleosome-wrapped CREs are refractory to TF binding, usually resulting in a lower affinity of TFs to the CRE; on the other hand, in animals, many pioneering TFs bind to a degenerate motif at nucleosomal targets. Such relaxation in binding is proposed to dramatically increase the searching space and ensure robust target access to variations in chromatin configuration [40–42]. We found that SCRM and MUTE mimic this canonical behavior of animal pioneer factors, as they displayed higher affinity towards nonnucleosomal than towards nucleosome-bound targets in ChIP-seq data (Fig 3C). In addition, nucleosome-bound SCRM and MUTE showed lower motif stringency at their binding sites (Fig 3B); this manifests as a slightly lower percent of nucleosome-bound peaks harboring the motif (50.3% versus 55.46% for SCRM and 26.8% versus 33.49% for MUTE). FAMA behaved differently from the other 2 bHLHs. It showed comparable affinity to its nucleosome-enriched and nonnucleosomal CREs (Fig 3C), and the motifs enriched at nucleosome-occupied binding sites of FAMA are more stringent than those at nonnucleosomal DNA binding sites (89.47% versus 69.3%).

To further understand the interaction between plant nucleosome-binding TFs and the genome, we characterized the genomic feature of these nucleosomal CREs. We found that unlike animal nucleosome-binding TFs, which function primarily at distal enhancers [43–46], MUTE, FAMA, SCRM, and LFY bind to nucleosomal CREs that are enriched in gene bodies (S4 Fig), possibly because the small size of the *Arabidopsis* genome contributes to the clustering and compaction of regulatory features, resulting in frequent overlap between enhancers and intronic regions [47,48]. Therefore, in *Arabidopsis*, precise recognition of an exact motif, instead of relaxed binding at nucleosomal targets, may be required for TFs to compete with other regulatory factors.

We next tested whether the stomatal lineage bHLHs have the potential to remodel nucleosomes by assaying changes in nucleosome occupancy at target sites in response to bHLH expression. Because MUTE resembles canonical animal pioneer factors, whose targets exhibited a particularly strong nucleosomal signal, and inducible expression of MUTE leads to large-scale and coordinated reprogramming of all epidermal cells to GC fate [49], we focused on MUTE. We assayed the profile of mononucleosomes by MNase-seq 4 hours post-estradiol induction. This time point was selected to ensure sufficient accumulation of MUTE protein

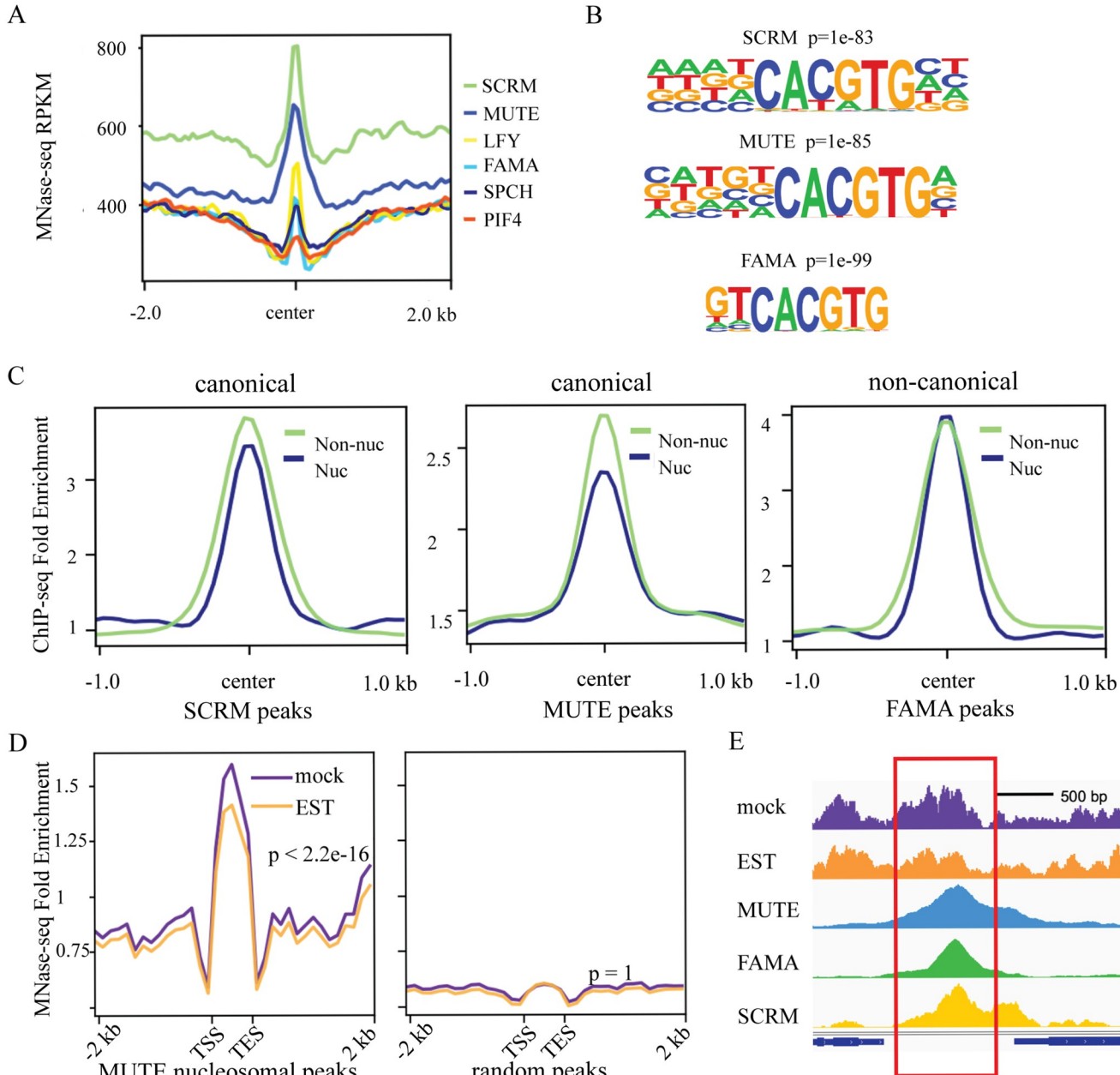

**Fig 3. The stomatal lineage bHLHs have the potential to bind nucleosomal DNA and induce chromatin remodeling. (A)** Nucleosomal density around the binding sites of the known pioneer factor LFY [18], the stomatal lineage bHLHs, and an unrelated bHLH, PIF4, in leaf tissue. Nucleosome density was determined by MNase-seq [38]. TF binding sites were determined from ChIP-seq data described in S1 Data. **(B)** Top nucleosome-associated ChIP-seq motifs identified at SCRM (top), MUTE (middle), and FAMA (bottom) peaks. **(C)** Average ChIP-seq fold enrichment around the center of Nuc or Non-nuc SCRM, MUTE, and FAMA peaks. MUTE and SCRM show a pattern similar to canonical animal pioneer factors. FAMA shows a noncanonical pattern that is different from animal pioneer factors. **(D)** Changes in nucleosome occupancy (MNase-seq signal) at nucleosomal MUTE peaks (left) and random peaks (right) after ectopic, EST-induced, MUTE expression. **(E)** Example of a decrease in nucleosome signal at a MUTE binding site upon MUTE induction. The top 2 traces represent the MNase-seq signal (nucleosome occupancy) after mock treatment (purple) and MUTE induction (EST, orange). The bottom traces show ChIP-seq binding of MUTE, FAMA, and SCRM at this locus. The data underlying this figure can be found in S1 Data. bHLH, basic helix–loop–helix; ChIP-seq, chromatin immunoprecipitation followed by deep sequencing; Non-nuc, nonnucleosomal; Nuc, nucleosomal; SCRM, SCREAM; TF, transcription factor.

while minimizing changes from downstream effectors that are not directly regulated by MUTE. Nucleosome positioning analysis revealed that the induction of MUTE indeed results in reduced nucleosomal density at its targets compared to a mock treatment (Fig 3D and 3E). We also generated estradiol-inducible lines to test nucleosomal profiles at targets of FAMA and SCRM; however, we were unable to consistently induce ectopic expression of these 2 bHLH factors with the strength and temporal dynamics needed for an effective profiling experiment.

## bHLHs associate with the SWI/SNF complex and HAC1 to potentially reprogram chromatin

Animal pioneer TFs can use different mechanisms to open closed chromatin. Structural studies showed that they can form stable complexes with target DNA in a nucleosome, distorting and relaxing interactions between DNA and histones and evicting linker histones [17]. Some pioneer factors also recruit chromatin remodelers to open closed chromatin [17,18]. As SCRM binding targets have the strongest nucleosome density among stomatal bHLHs, and to investigate whether MUTE-SCRM and FAMA-SCRM dimers might associate with chromatin modifiers to reshape the chromatin landscape, we conducted TurboID (TbID)-based proximity labeling to identify proteins associated with SCRM in its native context. Comparison of the complementing SCRM-TbID line with the cell type–specific controls and untreated SCRM-TbID samples revealed an enrichment of multiple subunits of the SWI/SNF chromatin remodeling complex, including the core ATPase BRAHMA (BRM), a core bromodomain-containing protein, BRD1, 2 regulatory subunits SWI3C and SWI3D, and several accessory proteins that have previously been found as part of the plant SWI/SNF complexes (Tables 1 and S1 and Fig 4A). We further found an enrichment for the histone acetyl-transferase HAC1,

**Table 1. Chromatin remodelers and coactivators identified as putative partners of stomatal lineage bHLH dimers.**

| | Gene name | AGI |
|---|---|---|
| **SWI/SNF complex** | BRM/CHA2/CHR2 | AT2G46020 |
| | BRD1 | AT1G20670 |
| | SWI3C | AT1G21700 |
| | SWI3D/CHB3 | AT4G34430 |
| | SWP73B/BAF60/CHC1 | AT5G14170 |
| | BRIP2 | AT5G17510 |
| | SYS1 | AT5G07940 |
| | SYS3 | AT5G07980 |
| | LUH | AT2G32700 |
| **Other chromatin remodelers** | CHR4/PKR1 | AT5G44800 |
| **HATs** | HAC1/PCAT2 | AT1G79000 |
| | HAC5* | AT3G12980 |
| **SAGA complex** | SPT20/PHL | AT1G72390 |
| | TRA1a | AT2G17930 |
| | TAF12b | AT1G17440 |

Partners were identified by Turbo-ID based proximity labeling with *SCRMp::SCRM-TbID-Venus* expressed in a homozygous *scrm* mutant background.

*No proteotypic (unambiguous) peptides identified. Spectral counts and label-free quantification values for these proteins are found in S1 Table.

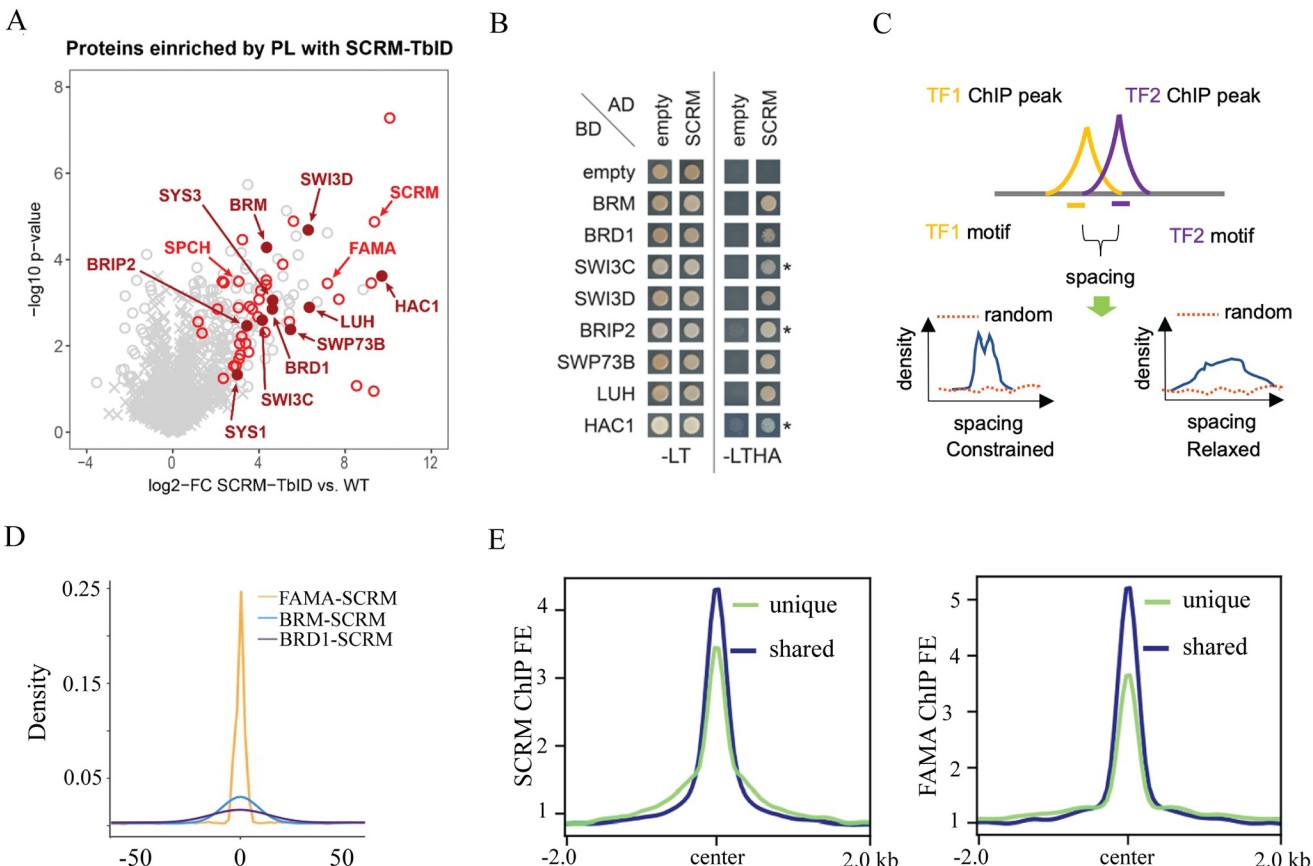

**Fig 4. The stomatal lineage bHLHs are associated with chromatin-related factors.** (A) Scatterplot of log2 fold enrichment of proteins in SCRM-TbID plants compared to wild-type samples as determined by unpaired 2-sided *t* test. Red circles are interaction candidates (proteins enriched vs. all controls). The positions of SCRM and its heterodimerization partners SPCH and FAMA are indicated in light red; positions of SWI/SNF components and HAC1 are indicated in dark red (filled circles). MUTE was not enriched due to the low abundance of the protein. (B) Y2H testing the interaction of SCRM (as AD-fusion) with the identified SWI/SNF complex components and HAC1 (as BD-fusion). Cells were spotted on SD-LT medium to confirm transformation with the constructs and SD-LTHA to test for interaction. Combinations with SWI3C, BRIP2, and HAC1 were grown on SD-LTHA with 5 mM 3-AT to suppress autoactivation (indicated by *). (C) Scheme of the spacing calculations between 2 DNA binding proteins used in D. Proteins that bind in a coordinated manner (blue line) show a narrow (constrained spacing) or wide (relaxed spacing) peak. Noncoordinated proteins (random, red dotted line) do not show a peak. (D) Density plot of the distance between SCRM and BRD1 (BRD1-SCRM), SCRM and BRM (BRM-SCRM), and SCRM and FAMA (FAMA-SCRM) shared ChIP-seq binding sites. Kolmogorov–Smirnov test: *p*-value for BRD1-SCRM vs. BRM-SCRM = 0.004787, other comparisons *p*-value < $2.2 \times 10^{-16}$. (E) Plots of average ChIP-seq signal fold enrichment at shared (dark blue) and unique (light green) SCRM (left) and FAMA (right) peaks. In both cases, peaks shared by both SCRM and FAMA had higher signals than unique peaks. The data underlying this figure can be found in S1 Data. AD, activation domain; BD, binding domain; bHLH, basic helix–loop–helix; ChIP-seq, chromatin immunoprecipitation followed by deep sequencing; SCRM, SCREAM; SPCH, SPEECHLESS; SWI/SNF, SWITCH DEFECTIVE/SUCROSE NONFERMENTABLE; TbID, TurboID; Y2H, yeast two-hybrid.

which we previously identified in a proximity labeling experiment with FAMA [50]. We confirmed the physical interaction between SCRM and several SWI/SNF complex components by yeast two-hybrid (Y2H) (Fig 4B). Interestingly, FAMA, but not MUTE, also shows strong interaction with many subunits of the SWI/SNF chromatin remodeler (S5A Fig). Interaction between the bHLHs and HAC1 was relatively weak in both Y2H (S5A Fig) and bimolecular fluorescence complementation (BiFC) assays (S5C Fig), suggesting that strong interaction with HAC1 may require the presence of the heterodimer or an additional linker protein (e.g., LUH, which interacts both with SCRM and FAMA and with HAC1 (S5A and S5B Fig). These data suggest that the FAMA-SCRM and MUTE-SCRM dimers could recruit the SWI/SNF

chromatin remodeler and the histone acetyl-transferase HAC1 to open local chromatin and activate transcription in stomatal development.

If these chromatin remodelers indeed function with SCRM, FAMA, and/or MUTE, we would expect them to be coexpressed and associated with the same genomic regions as the stomatal bHLHs. Analysis of publicly available scRNA-seq and ChIP-seq data for BRM and BRD1 [51] revealed that both are expressed in the stomatal lineage and bind to a large proportion of the stomatal lineage bHLH ChIP-seq peaks (S6A and S6B Fig). ChIP-seq data can also be used to determine the spacing and orientation of interacting factors on DNA, which can suggest the mode of action of the complex. Constrained complexes rely on simultaneous binding of each component with fixed orientation, as was found for the nucleosome-binding TF complex OCT4-SOX2, whereas relaxed conformations are seen for most TFs and their cofactors [52–54]. To determine the mode of interaction for the stomatal lineage bHLHs, we predicted the actual binding site with high affinity by calculating position weight matrices (PWMs) for the bHLHs, BRM, and BRD1 within 50 bp from their peak centers (Fig 4C). The positions of binding sites were then used to calculate the relative distances between each pair of interactors. The bHLH heterodimers follow a constrained spacing rule (Figs 4D and S6C) suggesting the heterodimers likely function as a single unit. Cooperative binding often leads to higher binding affinity [53], and we find that FAMA and SCRM exhibit higher binding affinity at shared sites than at their unique binding sites (Fig 4E). The interaction between the bHLHs and BRM or BRD1, however, is more relaxed (Figs 4D and S6C), suggesting extensive flexibility in binding and the existence of independent submodules that work together with the core bHLH complex [52]. It is noteworthy that the spacing between BRM and the bHLHs is more stringent than between BRD1 and the bHLHs (Figs 4D and S6C), which correlates with the relative strengths of physical interaction observed in the Y2H (Figs 4B and S5A). This is not due to lower binding affinity to chromatin, as BRM and BRD1 have similar level of ChIP-seq signal fold enrichment (S6D Fig). Together, these findings point to a pioneer factor-chromatin remodeler model where the bHLH heterodimers cooperatively bind to their targets as a single core unit, and the SWI/SNF complex acts as an "add-on" unit with BRM more tightly associated than BRD1.

## The SWI/SNF complex and HAC1 are required for GC differentiation

Because fate transitions in stomatal development are accompanied by changes in the chromatin landscape and the transcriptional master regulators associate with the SWI/SNF complex and HAC1, we designed artificial microRNAs (amiRNAs) [55] for the SWI/SNF genes *BRM* and *SWI3C* and for *HAC1* to knock down their expression in the stomatal lineage. For this analysis, we focused on the fate transition from GMC to terminally differentiated GCs promoted by FAMA because there are a number of clearly scorable phenotypes during this transition, and, after SCRM, FAMA showed the strongest interaction with HAC1 and SWI/SNF factors (S5 Fig). We found that driving *SWI/SNF* or *HAC1 amiRNA*s under the *FAMA* promoter resulted in GCs that divide again asymmetrically, often multiple times, and occasionally forming GCs or tumors inside of GCs (Figs 5A, 5B and S7). These abnormal GCs appeared as early as 5 days postgermination (5 dpg), with a progressive worsening of the phenotype over time. The control *pFAMA*-driven scrambled *amiRNA* did not cause any GC phenotypes (Fig 5A).

Morphological evidence suggested altered fate, so we assayed several molecular markers to better define the fate defects. ARABIDOPSIS THALIANA MERISTEM L1 LAYER (AtML1), an HD-ZIP class IV TF, is preferentially expressed in the epidermal layer and is required for epidermal cell differentiation [56]. *pAtML1* expression decreases in maturing tissue and a

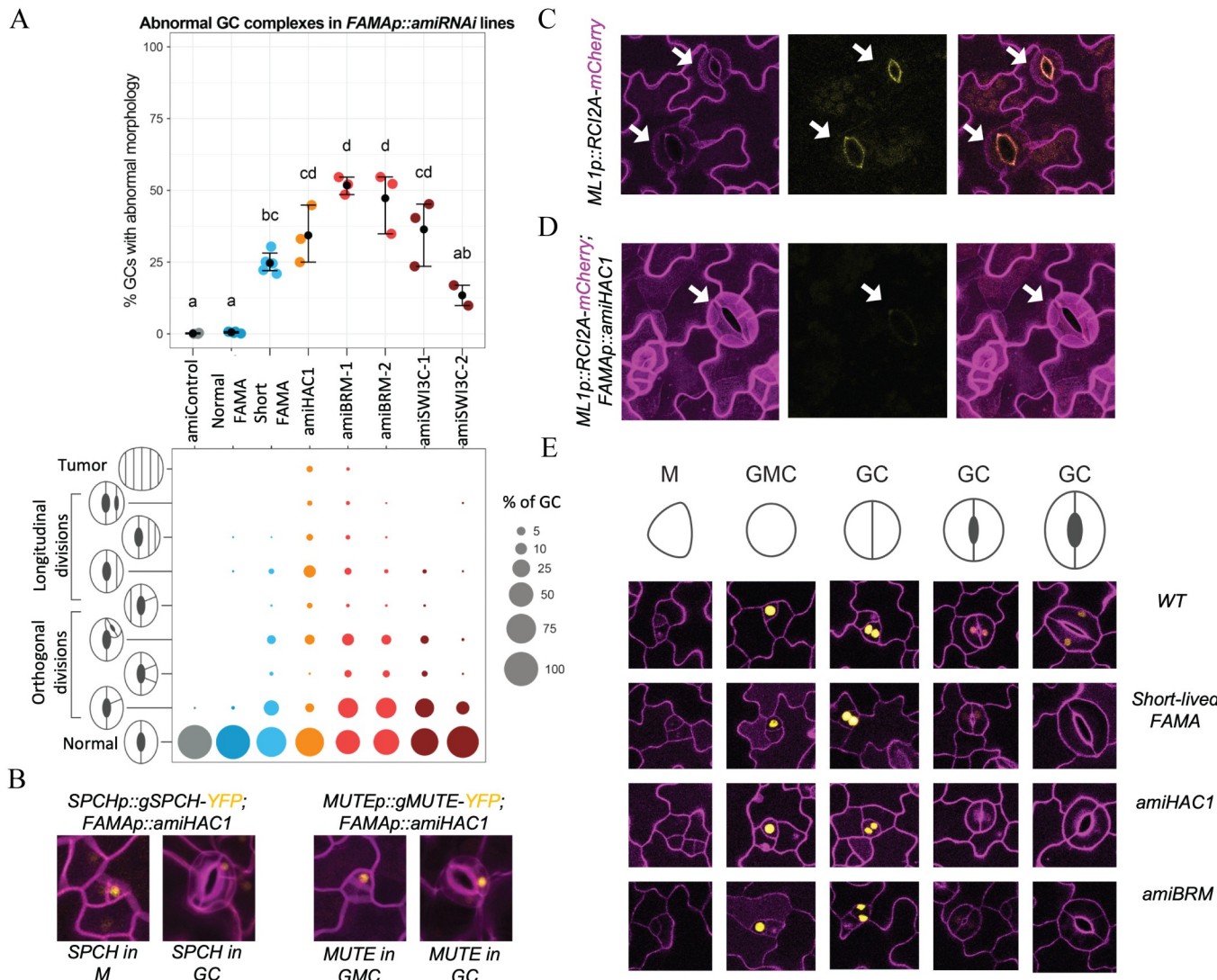

**Fig 5. The SWI/SNF complex and HAC1 are required for GC differentiation. (A)** Quantification of abnormal GC complexes in *FAMAp*::*amiRNA* lines of HAC1 (orange), BRM (red), and SWI3C (brown) compared to an amiRNAi control (scrambled sequence, grey) and to lines in which *fama* is rescued with a FAMA construct that persists to maturity (normal FAMA) and one with early-terminating expression ("short-lived" or short FAMA) in blue. The top panel reports the overall percentage of abnormal GCs in 21-day-old cotyledons. Each colored dot represents a plant with the mean and SE indicated as a black dot with whiskers. Statistical difference was tested by Kruskal–Wallis rank sum test (*p*-value = 0.0013), followed by a Conover post hoc test with holm correction (statistical difference between lines (adjusted *p*-value < 0.025) is shown as compact letter display above the data points). The bottom panel provides finer phenotypic dissection of abnormal GCs, with percentage of GCs in each of phenotypic classes represented by the size of the colored circles. **(B-E)** Confocal images of representative cells from cotyledon epidermis with cell outlines in magenta. **(B)** Reexpression of SPCH (left) and MUTE (right) translational reporters in GCs indicates reversion of GC cell identity to an early stomatal lineage state in *FAMAp*::*amiHAC1*. **(C, D)** Evidence that even morphologically normal *FAMAp*::*amiHAC1* GCs have fate defects. Compared to WT GCs **(C)** with pore autofluorescence (yellow) and decreased *ML1p*-driven reporter activity (magenta plasma membrane signal), in *FAMAp*::*amiHAC1* plants **(D)**, autofluorescence is missing and *ML1p* remains active in older GCs. Arrows point to morphologically mature stomata. **(E)** Comparison of FAMA expression during late stages of GC development, showing early termination in *FAMAp*::*amiHAC1* and *FAMAp*::*amiBRM*, similar to the "short-lived" FAMA line. The data underlying this figure can be found in S1 Data. BRM, BRAHMA; GC, guard cell; GMC, guard mother cell; M, meristemoid; SPCH, SPEECHLESS; SWI/SNF, SWITCH DEFECTIVE/SUCROSE NONFERMENTABLE; WT, wild-type.

reporter *ML1p*::*RCI2A-mCherry* becomes undetectable in wild-type mature GCs (Fig 5C). In GCs of the *pFAMA SWI/SNF or HAC1 amiRNAi* lines, however, ML1p::RCI2A-mCherry signal remains high in all GCs (Fig 5D), suggesting that they are not maturing normally. In addition, autofluorescence of the mature GC pore, which originates from chemicals such as ferulic

acid, p-coumaric acid, and cinnamic acid in the cell wall [57], is missing in *pFAMA SWI/SNF or HAC1 amiRNAi* lines (Fig 5C and 5D). Concomitant with failure to exhibit hallmarks for mature GC identity, the amiRNA lines inappropriately express early lineage markers such as SPCH and MUTE in the GCs (Fig 5B). Depletion of SWN/SNF components or HAC1, therefore, inhibits the full transition into mature GCs.

Insights into the mechanism by which SWI/SNF and HAC1 enforce GC fate commitment comes from the phenotype of a FAMA reporter line we termed "short-lived FAMA," in which FAMA protein expression was empirically shown to terminate early, before the GCs have fully matured. Short-lived FAMA produces a similar, albeit milder, phenotype as the *pFAMA SWI/ SNF* or *HAC1 amiRNAi* lines (Figs 5A and S7). We reasoned that if HAC1 or SWI/SNF were required to maintain *FAMA* expression in GCs, then introduction of the *amiRNA* constructs into a normally expressing FAMA translational reporter line (*pFAMA::gFAMA-YFP*) should cause premature disappearance of the FAMA-YFP signal. Indeed, these lines lost FAMA-YFP signal in very young GCs (Fig 5E), and quantitative reverse transcription PCR (qRT-PCR) measurement of *FAMA* transcript suggests that HAC1 and SWI/SNF regulate *FAMA* transcription (S8 Fig). *FAMA* is a potential target of MUTE, SCRM, and FAMA itself, and *FAMA* gene activation requires accessing CREs that are buried in relatively repressed chromatin regions (S8A Fig).

Our data suggest that the SWI/SNF complex and HAC1 maintain a transcriptionally active state of *FAMA* in maturing GCs, but it is less clear whether these factors play a role in the initial activation of *FAMA* in late GMCs. We do not see a delay in *FAMA* onset in the *pFAMA SWI/SNF* or *HAC1 amiRNAi* lines, but this may reflect the technical constraint that amiRNAs must first accumulate and deplete their targets. The fact that tumors, which are caused by a loss of *FAMA*, occasionally form inside the GCs in amiRNA lines (Figs 5A and S7) suggests that knocking down the SWI/SNF complex and *HAC1* can lead to a failure to properly induce *FAMA* expression. If this were the case, we would expect FAMA-SCRM nucleosomal target gene expression to also go down, and in RT-qPCR experiments, we found that maintenance of high-level expression of these targets also requires SWI/SNF chromatin remodelers and HAC1 (S8B Fig). These findings suggest the SWI/SNF complex and HAC1 function together with at least the FAMA-SCRM dimer to regulate gene expression and thus control GC fate commitment.

## Discussion

Cell type–specific transcriptional regulation is the driving force for cell identity establishment. Single-cell technologies provide exquisitely granular views of cellular heterogeneity and plasticity and afford opportunities to study how cell identity is established from a new perspective. In this work, by correlating scRNA-seq analysis with other genomic data, we began to define genome-wide regulatory mechanisms during stomatal fate transitions.

The compact nature of genomic DNA in the nucleus renders the chromatin environment around CREs intrinsically repressive in general [58], leading us to question how transcriptional regulation, especially activation, is initiated in closed chromatin regions during fate transitions. A well-established model in animals is that pioneer factors can target silent chromatin, bind to nucleosome-buried CREs, and launch transcriptome and epigenome reprogramming. Here, we provide evidence that the transcriptional master regulators of stomatal development have the potential for pioneering activity. We show that the MUTE-SCRM and FAMA-SCRM heterodimers bind to targets ensconced in repressive and closed chromatin, recruit the SWI/SNF ATPase-dependent chromatin remodelers and the histone

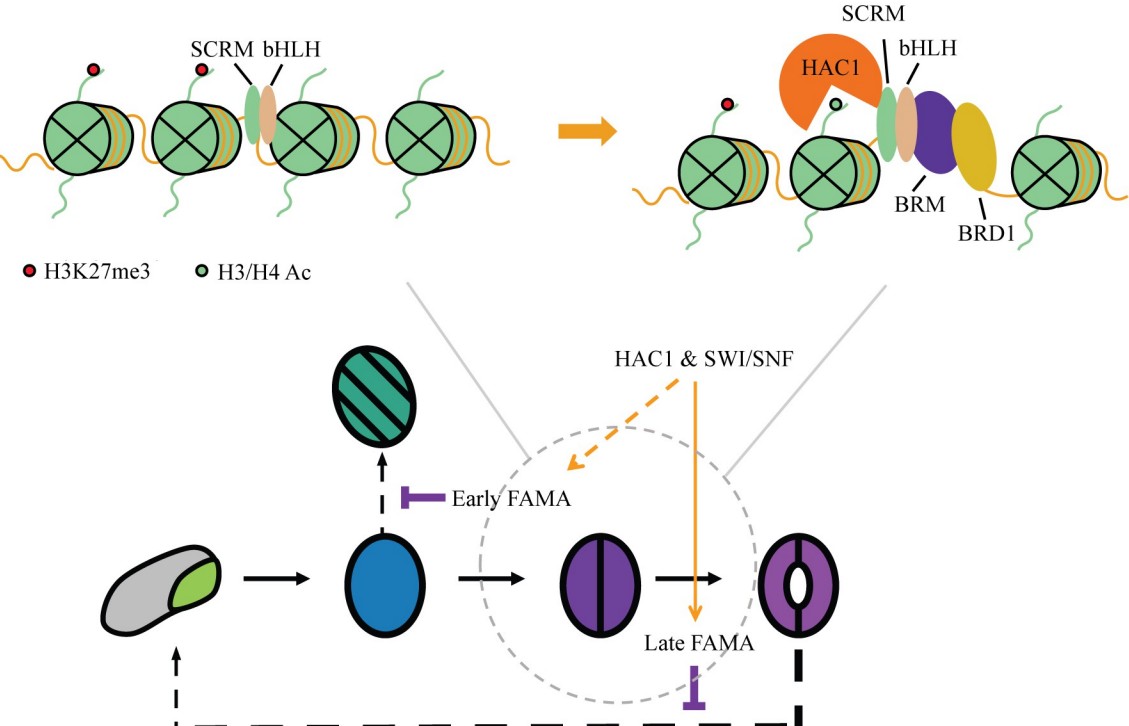

**Fig 6. A Model for chromatin reprogramming by the stomatal lineage bHLH heterodimers using transcriptional regulation of FAMA as an example.** (**Top**) The stomatal lineage bHLH TFs form heterodimers and can access targets located in repressive chromatin regions. They then recruit the SWI/SNF complex and HAC1 to evict nucleosomes and acetylate histone tails to induce chromatin reprogramming and maintain the expression of their targets at different stages of GC fate commitment. (**Bottom**) HAC1 and the SWI/SNF complex could impact FAMA and FAMA target gene expression at 2 stages. Experimental evidence suggests that FAMA itself is kept active by the action of HAC1 and the SWI/SNF complex (late FAMA), and this enables FAMA to promote terminal differentiation. Based on indirect evidence, early FAMA expression, which enables it to repress cell cycle progression, could also depend on HAC1 and SWI/SNF activity. bHLH, basic helix–loop–helix; GC, guard cell; SWI/SNF, SWITCH DEFECTIVE/ SUCROSE NONFERMENTABLE; TF, transcription factor.

acetyltransferase HAC1 to open local chromatin, and activate transcription and thus control cell fate transitions (Model in Fig 6).

MUTE and SCRM behave like canonical animal pioneer factors and exhibit higher affinity toward nonnucleosomal targets. FAMA, however, showed comparable affinity to its nucleosome-enriched and nonnucleosomal CREs and higher motif enrichment in nucleosomal regions. The opposing motif enrichment behavior of SCRM and FAMA is reminiscent of the pioneer factor pair FoxA2 and GATA4, which act together at a specific stage during animal cell differentiation. While the FoxA2 motif is enriched at nonnucleosomal targets compared to nucleosomal targets of the pair, the opposite can be observed for the GATA motif, indicating that motif preferences are altered at cobinding sites [42]. Together with the fact that FAMA has a relatively low mononucleosome signal at its targets and nucleosomal FAMA peaks are significantly ($p = 0.00660$) more enriched in SCRM binding, this could indicate that FAMA requires SCRM to bind nucleosomal DNA and may not have chromatin binding activity on its own.

While the stomatal lineage bHLHs share common features with animal pioneer factors, they are unusual in the sense that the MUTE-SCRM and FAMA-SCRM heterodimers function together as a single unit for which the binding affinity at nucleosome-occupied regions is high. Interestingly, heterodimerization of the animal bHLH-type pioneering factor ASCL1 with

E12α also greatly increases its affinity for nucleosome binding [59]. Structural modeling of DNA-bound stomatal lineage bHLH dimers revealed a similar structure to that of ASCL1 (S9 Fig). Both the animal and the stomatal lineage bHLH dimers form a typical scissor-like helix–loop–helix structure with the first helix pair enclosing the DNA. Importantly, though, the DNA binding helix is short, compared to other nonpioneer bHLHs, and does not extend past the DNA and would therefore not interfere with a histone molecule binding on the opposing side [41]. The similarities in binding affinity and structure between the SCRM and ASCL1 heterodimers suggest that pioneering bHLH dimers may bind to nucleosomes in a conserved manner. This is particularly interesting because ASCL1 is expressed transiently in neural progenitor cells to induce neural differentiation, a process similar to stomatal development where a transition from cell division to differentiation is required. Notably, ASCL1 acts as a pioneer factor both by itself (as a homodimer), to promote neurogenesis, and as a heterodimer with the ubiquitously expressed E12α, to reprogram fibroblasts to mature into neuronal cells [41,59]. Similar to E12α, SCRM is expressed throughout the stomatal lineage. This opens the question whether the repressive chromatin association features we identified for SCRM are a result of heterodimerization with MUTE and FAMA, or vice versa, and whether the 3 of them can overcome repressive chromatin barriers by themselves. Although these bHLHs likely do not form homodimers under normal circumstances, one could try to force homodimerization of individual bHLHs and examine their affinity to nucleosomal DNA to answer this question.

Specification and maturation of guard cells require extensive changes to chromatin and gene expression. FAMA is likely the major facilitator of those changes. FAMA is expressed from the late GMC stage through to GC maturity. Loss of FAMA abolishes GC production and leads to the formation of epidermal tumors caused by repeated symmetric divisions of the GMC with almost parallel division planes [24]. The phenotype of the "short-lived FAMA" line (i.e., occasional reinitiation of ACDs and the formation of GCs within mature-looking GCs) suggests termination of the GMC cell division program and GC fate commitment are 2 separate functions of FAMA (Fig 6). Although a pulse of FAMA expression is sufficient to repress excessive cell division, prolonged FAMA expression is required for GCs to complete differentiation. That FAMA-promoter-driven amiRNAs targeting HAC1 or the SWI/SNF complex result in loss of late FAMA expression suggests that chromatin remodelers are needed for sustained FAMA expression. For technical reasons related to the time required for amiRNAs to eliminate their targets, we cannot conclude that the initial high level of FAMA protein in these amiRNA experiments means that the SWI/SNF complex and HAC1 are not required for the initial activation of FAMA. In fact, we occasionally observe the formation of "*fama* tumors" within GCs in some of the lines. This could be caused by a persistence of the effect of reduced SWI/SNF components or HAC1 on the *FAMA* locus, which, in turn, causes a failure to activate *FAMA* when the cell reenters the stomatal lineage.

FAMA acts primarily as transcriptional activator [24,35,36], which fits well with the proposed model where the FAMA-SCRM heterodimer binds to regulatory regions of silent genes that reside in repressive chromatin to activate transcription. Disruption of the FAMA-RBR interaction, and presumably recruitment of the PRC2 complex, also leads to a reinitiation of ACDs [24], suggesting that both transcription activation and epigenetic silencing are vital for GC fate commitment by FAMA and that FAMA may be involved in all the stages of GC fate commitment where we see opposite trends in transcriptional entropy and the number of genes expressed. Both activation and repression machineries appear to be involved in fate specification, raising many questions about how they interact to ensure proper regulation at each target locus. The SWI/SNF complex has been shown to repel the PRC2 complex [60]. Additional studies may reveal how these complexes interact with FAMA and each other to ensure proper regulation at each target locus.

We provide compelling evidence that MUTE-SCRM and FAMA-SCRM complexes can access repressive chromatin and promote remodeling. Whether SPCH has such potential is less clear. It is attractive to think that as the initiator of the stomatal lineage, SPCH would participate in chromatin remodeling, but lineage pioneering need not be the same as biochemical pioneering. With our current datasets, however, we cannot be certain whether the equivocal results for SPCH do not just reflect technical limitations. There are several challenges to assaying SPCH, the first being that it expressed quite early and in a population of cells with multiple potential fates, making it tricky to obtain transcriptionally homogeneous populations of cells. In addition, unlike ectopic expression of MUTE, which induces strong and synchronized appearance of guard cell features, overexpression of SPCH causes excessive cell divisions without driving cells toward a unique identity. Continual improvement in the sensitivity of single-cell technologies, however, may enable questions about SPCH association with chromatin to be addressed in the future.

Taken together, this study describes how TFs potentially access repressive chromatin and induce chromatin remodeling during a normal developmental process in plants. Our data confirm the involvement of closed, chromatin-buried distal intergenic regions and regions within gene bodies in transcriptional regulation.

Combinatorial and integrated genome-wide comparisons of the behaviors of related TFs proved to be a powerful way to reveal trends that may not have been clear from analysis of any of these factors alone. Layering on single cell type or single cell transcriptome and epigenome profiles improved the resolution of our analysis, and there are prospects for even finer resolution as more single-cell datasets are being produced by the community. Our data and our approaches will serve as a starting point for future studies on transcriptional and epigenetic regulation during cell fate transition and reprogramming.

## Materials and methods

### Materials availability

Plasmids and seed generated in this study have been deposited in relevant collections of the Bergmann lab at Stanford University and will be distributed upon request.

### Plant materials and growth conditions

All plants in this study were germinated and grown on half-strength Murashige and Skoog (1/2MS) agar (0.8%) plates under standard growth conditions (21˚C,16-hour light/8-hour dark) in a Percival growth chamber for 3, 5, 7, 10, or 21 days, according to the experimental purposes. The following plant materials were used in this study (all lines are ecotype Col-0): Col-0 wild type (WT), *ESTpro::MUTE* [49], *fama-1* [24], *MUTEpro::gMUTE-YFP* [49], *FAMApro::FAMA^LGK^/fama-1* [36], *FAMApro::FAMA-MYC/fama-1* [36], *FAMApro::gFAMA-YFP/fama-1* [61] and short-lived *FAMApro::gFAMA-YFP/fama-1* (described below), *ML1pro: mCherry-RCI2A* [62], *ML1pro::YFP-RCI2A* [62], *mute* [23], *SPCHpro::gSPCH-YFP/spch-3* [27], *SCRMp::SCRM-TbID-Venus/ice1-2* (described below), *SPCHp::TbID-YFPnls* (described below), *FAMAp::TbID-YFPnls* [50].

### Plasmid construction and plant transformation

The constructs described here were created using Gateway LR recombination (Invitrogen 12538120). For *FAMApro::amiRNA* constructs, artificial microRNA sequences were designed for control miRNA, *HAC1*, *BRM*, or SWI3C with WMD3 (http://wmd3.weigelworld.org) and amplified through the website's standard protocol using the pRS300 plasmid as a template.

*pDONR-L4-FAMApro-R1* [61], *pENTR_L1-amiRNAi-L2*, and R4pGWB501 or R4pGWB601 [63] were recombined. Constructs were transformed into the plasma membrane marker expressing lines *ML1pro::mCherry-RCI2A* or *ML1pro::YFP-RCI2A* [62]. The short-lived FAMA line is an individual line that harbors the same construct as *FAMApro::gFAMA-YFP/ fama-1* [61] but expresses *gFAMA-YFP* for a shorter period of time in each developing GC. The *SCRMp::SCRM-TbID-Venus* construct was generated by recombining *pENTR 5′-L4-SCRMpro-R1* (containing a 2,579-bp upstream fragment), *pENTR/D-L1-SCRM-L2* (containing SCRM amplified from *Arabidopsis* cDNA), *pDONR R2-Turbo-mVenus-STOP-R3* [50], and *pB7m34GW0* [64] and transformed into *ice1-2 +/−* (SALK_00315) plants. Lines homozygous for the transgene and *ice1-2* were selected in T2 and T3. *SPCHp::TbID-YFPnls* was recombined from *pENTR-L4-SPCHpro-R1* [64], *pENTR L1-Turbo-YFP-NLS-STOP-L2* [50], and *R4pGWB60* [63] and transformed into Col-0 WT. All constructs were transformed by floral dipping [65] using the *Agrobacterium* strain GV3101. Primers used for cloning and genotyping are described in S2 Table.

## ChIP-seq

MOBE ChIP on FAMA was carried out as previously described [66]. Briefly, 25 g of 4-day-old whole seedlings of *FAMApro::FAMA-MYC/fama-1* and Col-0 (WT control) were used. Samples were processed in standard-sized aliquots during crosslinking, nuclei isolation, and DNA fragmentation steps before combining for immunoprecipitation. Chromatin was fragmented by a Bioruptor (Diagenode) programed at high intensity for $3 \times 7.5$ minutes (cycles of 30 seconds on and 30 seconds off) at 4˚C. Immunoprecipitation was performed with a monoclonal anti-MYC antibody (71D10; Cell Signaling Technology), followed by incubation with magnetic beads (Dynabeads Protein A; Invitrogen). Immunoprecipitated DNA was purified by the ChIP DNA Clean & Concentrator (Zymo). Libraries were constructed using the Nugen Ovation Ultralow Library System following the manufacturer's protocol. Col ChIP DNA was used as a control. Libraries were sequenced 50 bp with single end on an Illumina HiSeq 2000. LFY, SPCH, MUTE, SCRM, PIF4, BRM, BRD1, H3K27me3, and H3K4me3 ChIP-seq datasets were retrieved from GEO (dataset identifiers provided in S1 Data). All previously published ChIP-seq experiments except LFY were performed in seedlings grown in comparable conditions at similar stages to the FAMA ChIP experiment described above. LFY ChIP was performed in root explants.

## MNase-seq

Approximately 3.5 dpg *ESTpro::MUTE* seedlings were submerged in 2 μM estradiol or ½ MS liquid media as a control for 4 hours. Then, 0.5 g of seedlings were harvest and ground in liquid nitrogen. Nuclei were prepared as described previously [38] with the following modifications. The isolated nuclei were washed twice in 1 ml of HBB buffer (25 mM Tris-Cl (pH 7.6), 0.44 M sucrose, 10 mM MgCl2, 0.1% Triton-X, and 10 mM beta-mercaptoethanol). Nuclei were then treated with 0.5 U/μl final concentration of Micrococcal Nuclease (NEB) in digestion buffer (16 mM Tris-Cl (pH 7.6), 50 mM NaCl, 2.5 mM CaCl2, 0.01 mM PMSF, and EDTA-free protease inhibitor cocktail (Roche)) for 3 minutes at 37˚C. Digestion was stopped with 10 mM EDTA and treated with Proteinase K (Roche). DNA was purified with phenol:chloroform:isoamyl alcohol (25:24:1) extraction and precipitated with 1/10 volume of 5 M sodium acetate and 2 volumes of ethanol. Purified DNA was run on 2% agarose gel, and bands corresponding to approximately 150 bp were cut and purified with a Gel Purification kit (Promega). Libraries were constructed using the Nugen Ovation Ultralow Library System V2 following the manufacturer's protocol. Approximately 75 bp of pair-end reads were sequenced on NextSeq 4000.

## NGS analysis

ChIP-seq reads were mapped to the TAIR10 [67] *Arabidopsis* genome using bowtie2 [68]. PCR duplicates were removed by SAMtools [69]. Uniquely mapped reads were retained for peak calling using the MACS2 [31] and for metaplots. Each ChIP contained at least 2 biological replicates. MACS2 was run using default settings and peaks with an FDR cutoff of $10^{-6}$ over Col control were selected. De novo motif discovery was conducted using the findMotifsGenome function of HOMER [70] with sequences that are 200 bp surrounding the summits of peaks. Genes were annotated by ChIPseeker32 with a 3-kb cutoff. Reads pile-up were calculated by BEDtools [71], and fold changes were calculated by DEseq2 [72].

scRNA-seq datasets were analyzed using standard Seurat [73] pipeline with the same parameters as described in [27]. Entropy calculation for scRNA-seq uses the standard Shannon entropy form [13]:

$$S = -\sum_i Pi \times \log(Pi)$$

where Pi represents the probability of expressing a given gene *i* in a cell. The entropy can be computed by dividing the number of counts for gene *i* by all of the gene counts in a given cell [13].

The stomatal bHLHs' targets were determined by incorporating scRNA-seq datasets with the ChIPseq data sets. A list of targets is included in S1 Data and can be found in the sheet labelled bHLHs' targets. Specifically, genes that are expressed in at least 25% of cells that express specific stomatal bHLHs with a log2(fold change) of at least 0.223 were assigned as targets if they have at least 1 binding site of the corresponding bHLH. Heat maps and metaplot profiles of relative enrichment (log2 ChIP/Input) at defined genomic regions were generated by deeptools [74,75]. BigWig files of the mapped reads were visualized by the Integrative Genomics Viewer (IGV) [76].

Data processing for MNase-seq is similar to ChIP-seq, except (1) mapping is performed with pair-end mode, and (2) DANPOS [77] was used to identify nucleosome occupied regions using $q < 10^{-50}$ as cutoff. ATAC-seq datasets in different types of tissue were retrieved from published data (see S1 Data). Data processing for ATAC-seq is similar to ChIP-seq, except (1) mapping is performed with pair-end mode, and (2) HOTSPOT [78] was used to identify accessible regions.

## PL-MS

Proximity labeling with 3 replicates per line and condition was performed as previously described [50] with minor modifications. Briefly, 5-day-old Col-0 WT, *SCRMp::SCRM-TbID-Venus*, *SPCHp::TbID-YFPnls*, and *FAMAp::TbID-YFPnls* seedlings, grown on ½ MS plates with 0.5% sucrose, were submerged in liquid ½ MS medium with or without 50 μM biotin for 2 and 2.5 hours, respectively. Proteins were extracted from approximately 1.3 g plant material with buffer 1 (50 mM Tris (pH 7.5), 150 mM NaCl, 0.1% SDS, 1% NP-40, 0.5% Na-deoxycholate, 1 mM EDTA, 1 mM EGTA, 1 mM DTT, 20 μM MG-132, 1 × complete proteasome inhibitor, 1 mM PMSF), cleared from debris by centrifugation, and free biotin was depleted by buffer exchange to buffer 2 (50 mM Tris (pH 7.5), 150 mM NaCl, 0.1% SDS, 1% NP-40, 0.5% Na-deoxycholate) using PD-10 desalting columns (GE Healthcare). Around 14 mg protein were used for pulldowns with 200 μl streptavidin beads (Dynabeads MyOne Streptavidine T1, Invitrogen; 0.5 × complete and 0.5 mM PMSF added). Beads were washed twice with cold buffer 2, twice each with cold buffer 2 containing 500 mM and 1 M NaCl, once each with buffer 3 (2 M Urea, 1% SDS in 10 mM, 50 mM Tris (pH 7.5)) and 4 (2 M Urea, 50 mM Tris

(pH 7.5)), once with cold 50 mM Tris (pH 7.5) and stored at −80˚C. For on-beads digest, beads were washed again with 50 mM Tris (pH 7.5) and once with 50 mM Tris (pH 7.5), 1 M Urea. Trypsin digest and desalting and LC-MS were done as previously described. Protein identification and label-free quantification (LFQ) were done with MaxQuant (version 1.6.8.0) [79] with standard settings, fast LFQ disabled, and an LFQ min. ratio count of 1, searching against the TAIR10 database (TAIR10_pep_20101214, updated 2011-08-23, www.arabidopsis.org). Data filtering and statistical analysis were done in Perseus (version 1.6.2.3) [80]. LFQ values were log2 transformed and missing values were imputed from normal distribution using total matrix mode and standard settings. SCRM-TbID-enriched proteins were identified by consecutive pairwise comparisons with the controls (2-sided $t$ tests with permutation-based FDR of 0.05 or 0.1 and S0 of 0.5, considering only proteins identified in at least 2 replicates of biotin-treated SCRM-TbID). Proteins enriched versus WT (FDR = 0.05) were further tested for enrichment versus *SPCHp*- and *FAMAp*-driven nuclear TbID and untreated SCRM-TbID samples (FDR = 0.1). To compensate for higher expression of TbID by the FAMA promoter, protein LFQs values were normalized to TbID in each sample for this comparison. Proteins enriched versus all controls and identified by MS/MS were considered high-confidence candidates.

## Y2H

The full-length coding sequences of *SCRM*, *SCRM2*, *SPCH*, *MUTE*, *FAMA*, *BRM*, *BRD1*, *SWI3C*, *SWI3D*, *BRIP2*, *SWP73B*, *LUH*, and *HAC1* were amplified from *Arabidopsis thaliana* cDNA and cloned into pGAD-T7 or pGBK-T7 (Clontech). Combinations of bait and prey plasmids were then cotransformed into yeast strain AH109, and positive transformants were selected for by growth on synthetic dropout (SD) medium without leucine and tryptophan (−LT). Bait–prey interaction was tested by growth complementation assays on SD medium without leucine, tryptophan, histidine and adenine (−LTHA) as described in the Matchmaker GAL4 Two-Hybrid System 3 manual (Clontech). To overcome autoactivation from some of the constructs, 3-amino-1,2,4-triazole (3-AT) was added to nutrient selection plates at different concentrations.

## BiFC

Coding sequences of *SCRM*, *MUTE*, *FAMA*, and *HAC1* were first cloned into pENTR/D-TOPO using TOPO cloning and then subcloned into pDEST-GW-VYCE and pDEST-VYCE(R)-GW for C- and N-terminal YFP-C fusions (*FAMA*, *HAC1*) or into pDEST-GW-VYNE and pDEST-VYNE(R)-GW for C- and N-terminal YFP-N fusions (*SCRM*, *MUTE*, *FAMA*) using Gateway LR recombination [81]. Plasmids were transformed into Agrobacteria (strain GV3101). For BiFC, Agrobacteria from an overnight culture were diluted in LB medium containing antibiotics and grown for 2 hours, pelleted, resuspended in LB medium supplemented with 150 μM Acetosyringone and grown for another 4 hours at 30˚C, pelleted again and resuspended in a 5% sucrose solution to an OD600 of 2. A YFP-C and a YFP-N fusion-containing suspension were mixed with a suspension of Agrobacteria containing a 35S::p19 plasmid (tomato bushy stunt virus (TBSV) protein; suppresses silencing) at a 1:1:0.5 ratio and infiltrated into young *Nicotiana benthamiana* leaves. Three leaves were infiltrated per protein pair to test, with all 4 possible combinations infiltrated on different quadrants of the same leaf for improved comparability. Two days after infiltration, leaf sections were imaged on a Leica Stellaris confocal microscope with HyD detectors using a 25× water objective, retaining the same settings for all images. Images were visualized using fire LUT (Look

Up Table) in Fiji [82] setting display range for brightness from 0 to 100 or 0 to 1,000 for all images.

## Microscopy and phenotypic analysis

Confocal images were taken with a Leica SP5 or Stellaris confocal microscope. Cell outlines were visualized by propidium iodide staining (0.01 mg/ml in water, Molecular Probes P3566) or the plasma membrane markers *ML1pro*::*mCherry-RCI2A* or *ML1p*::*YFP-RCI2A* [62]. *ML1* promoter activity was visualized by *ML1pro*::*mCherry-RCI2A*. For GC phenotype quantification, cotyledons of 21 dpg plants were cleared in 7:1 ethanol:acetic acid solution and mounted in Hoyer's solution. DIC images of the entire cotyledon were taken with a Leica DMi8 inverted scope using a 20× objective and the tiling function. All visible GCs on the abaxial side of the cotyledons were counted and classified. For each line, between 2 and 5 cotyledons were scored. Detailed images of GCs were taken with a 40× objective. All images were processed in ImageJ/Fiji [82].

## RNA extraction and RT-qPCR

Around 50 mg of true leaves for each replicate and 3 replicates were harvested from Col-0, *FAMApro*::*amiHAC1*, and *FAMApro*::*amiBRM* plants 14 dpg. RNA was extracted using the RNeasy Mini kit (Qiagen 74004) coupled with DNase I treatment (Qiagen 79254) following the manufacturer's instructions. For each replicate, 2 ug RNA was used to synthesize cDNA with the iScript cDNA Synthesis Kit (Bio-Rad 1708890). The SsoAdvanced Universal SYBR Green Supermix (Bio-Rad 1725271) was used for qPCR, which was performed on a CFX96 Real Time System (Bio-Rad).

## 3D structure predictions

The 3D crystal structure of the nucleosome was retrieved from the protein data bank (https://www.rcsb.org/; PDB: 1kx5). The structures of bHLH dimers were modelled onto existing 3D structures of DNA-bound bHLH dimers using SWISS-model: HsMyoG was modelled on mouse MyoD (PDB: 1mdy) and the HsAsclI-HsTcf3(E12a) dimer was modelled on mouse NeuroD1-Tcf3 (PDB: 2ql2) as previously published [41]. For the SCRM dimers, multiple models were built and the model with the best score (QMEANDicCo Global), human Mad-Max (PDB: 1nlw), was chosen for display. The structures were visualized in PyMOL (https://pymol.org/2/). Modelled dimers are shown with the DNA of their model template. Because helix length varied slightly between different models based on the template, helix length in the selected SWISS-model models were further compared to helix predictions from PSIPRED 4.0 (http://bioinf.cs.ucl.ac.uk/psipred) [83] and AlphaFold [84,85] (https://alphafold.ebi.ac.uk/; structure database numbers: A0A178UF96, A0A178VCY0, A0A178VKX8, A0A384KCX7).

## Supporting information

**S1 Fig. Transcriptional entropy during differentiation of root cell types. (A)** Entropy scores (left) and cell identities (right) in root cells derived from scRNA-seq dataset in [26]. **(B)** Boxplot of entropy scores of different root cell types along a developmental gradient where meristem represents the least differentiated cells, followed by maturation and then elongation (most differentiated). The data underlying this figure can be found in S1 Data.
(TIF)

**S2 Fig. Shared properties among the stomatal lineage bHLHs' targets. (A)** A Venn diagram showing the overlap of ChIP-seq peaks among the stomatal bHLHs, data sources shown in S1

Data. **(B)** Top motifs in the stomatal bHLHs' binding sites as determined by ChIP-seq of the respective bHLHs. Note that all factors recognize a G-box motif, but SPCH has an additional alternative preferred binding site. bHLH, basic helix–loop–helix; ChIP-seq, chromatin immunoprecipitation followed by deep sequencing; SPCH, SPEECHLESS.
(TIF)

**S3 Fig. Chromatin landscapes at SPCH, MUTE, FAMA, and SCRM targets. (A)** FE of H3K4me3 levels in WT GC (purple) and FAMA^LGK GCs [37] ("pre-GCs" in orange) at targets of indicated stomatal bHLHs. **(B)** Chromatin accessibility of stomatal bHLHs' binding sites and randomly sampled genomic intervals in mesophyll cells (top) and in whole seedlings (bottom) [34]. The data underlying this figure can be found in S1 Data. bHLH, basic helix–loop–helix; FE, fold-enrichment; GC, guard cell; SCRM, SCREAM; SPCH, SPEECHLESS; WT, wild type.
(TIF)

**S4 Fig. Nucleosomal binding sites of plant TFs are enriched in the gene body.** Histograms showing the distribution of all targets (left) and nucleosomal targets (right) of LFY **(A)** SCRM **(B)** MUTE **(C)** and FAMA **(D)**. The red dot marks distal intergenic regions. The data underlying this figure can be found in S1 Data. LFY, LEAFY; SCRM, SCREAM; TF, transcription factor.
(TIF)

**S5 Fig. Interaction between chromatin factors and stomatal lineage bHLHs. (A)** Full panel of Y2H assays between AD-fused SCRM, SCRM2, SPCH, MUTE, and FAMA and BD-fused SWI/SNF components and HAC1. Yeast was spotted onto SD-LT to confirm cotransformation of the constructs and onto SD-LTHA to test for interaction. 3-AT was added to some SD-LTHA plates as indicated to overcome autoactivation of SWI3C, BRIP2, and HAC1 (indicated by x). Interactions with SCRM are also shown in Fig 3B. **(B)** Y2H of HAC1 and LUH. **(C)** FAMA and MUTE, but not SCRM, interact weakly with HAC1 via bimolecular fluorescence complementation (BiFC) in *N. benthamiana*. V-N: Venus N-terminal half, V-C: Venus C-terminal half. AD, activation domain; BD, binding domain; bHLH, basic helix–loop–helix; SCRM, SCREAM; SPCH, SPEECHLESS; SWI/SNF, SWITCH DEFECTIVE/SUCROSE NON-FERMENTABLE; Y2H, yeast two-hybrid; 3-AT, 3-amino-1,2,4-triazole.
(TIF)

**S6 Fig. *BRM*, *BRD1*, and *HAC1* are expressed in the stomatal lineage and the proteins bind to regions in the genome shared with the stomatal lineage bHLHs. (A)** Expression levels of *BRM*, *BRD1*, *HAC1*, and stomatal lineage markers in the scRNA-seq dataset from young developing true leaves. Cell type abbreviations are AF, alternative epidermal (likely pavement cell) fate; M, meristemoid; GC, guard cell. **(B)** A Venn diagram showing the overlap of ChIP-seq peaks of BRM, BRD1, and SCRM. **(C)** Cumulative plot of the distributions of spacing between indicated TF pairs shown in Fig 4D. **(D)** Average ChIP-seq signal (count per million reads) at BRD1 and BRM peaks that overlap with SCRM (blue) and those that do not (green). The data underlying this figure can be found in S1 Data. bHLH, basic helix–loop–helix; BRM, BRAHMA; ChIP-seq, chromatin immunoprecipitation followed by deep sequencing; SCRM, SCREAM; scRNA-seq, single-cell RNA-sequencing; TF, transcription factor.
(TIF)

**S7 Fig. *FAMAp::amiRNA* lines display diverse developmental defects in GCs. Top**: GC variations observed in the lines described in Fig 5A are shown as simplified schemes in the top panel. **Left**: Examples for each class of GCs. Image borders are color-coded to match the top

scheme. Transverse and longitudinal extra divisions in GCs are marked by filled and unfilled arrowheads, respectively. GCs within GCs are marked by an asterisk. **Right**: Zoomed out example images of the abaxial epidermis of the indicated lines showing the distribution of GCs with normal and abnormal morphology (overlays using the same color code as the top scheme). The images are DIC images of cleared 21-day-old cotyledons. Scale bar = 100 μm. (TIF)

**S8 Fig. *FAMAp::amiRNA* lines fail to up-regulate FAMA targets. (A)** View of the FAMA genomic locus, repressive H3K27me3 signal in "pre-GCs" (LGK, light blue) and GCs (green) and binding of BRM (purple) and the stomatal bHLHs (SPCH, light blue; MUTE, dark blue; FAMA, green; SCRM, orange) from ChIP-seq experiments. **(B)** Relative expression levels of SCRM-FAMA targets in *FAMAp::amiBRM* and *FAMApro::amiHAC1* lines as assayed by qRT-PCR; RNA extracted from 14-day-old leaves and normalized to actin. The data underlying this figure can be found in S1 Data. bHLH, basic helix–loop–helix; BRM, BRAHMA; ChIP-seq, chromatin immunoprecipitation followed by deep sequencing; GC, guard cell; qRT-PCR, quantitative reverse transcription PCR; SCRM, SCREAM; SPCH, SPEECHLESS. (TIF)

**S9 Fig. Predicted structure of DNA-bound SCRM dimers is compatible with nucleosome binding. (A)** X-ray crystal structure of the nucleosome core particle at 1.9 A resolution (PDB: 1kx5). The DNA and histone octamer are shown in grey and dark red, respectively. **(B)** Scheme of nucleosome-compatible and nucleosome-incompatible bHLH dimers. Histones do not interfere with binding of short DNA binding helices on the other face of the DNA (left), while binding of long DNA binding helices is obstructed (right). **(C, D)** Examples of human pioneer (Ascl1-Tcf3/E12a, **C**) and nonpioneer (MyoG, **D**) bHLH dimers, modelled with SWISS-model. The end of the DNA binding helix (dark green) is marked with an arrowhead. **(E-H)** 3D structures of plant bHLH heterodimers modelled with SWISS-model: PIF4 **(E)**, SCRM-SPCH **(F)**, SCRM-MUTE **(G)**, SCRM-FAMA **(H)**. The end of the DNA helices of the SCRM heterodimer is of similar length to the Ascl1-Tcf3 dimer and does not protrude beyond the DNA. The PIF4 heterodimer, in contrast resembles MyoG. **(I)** Comparison of the length of the DNA binding helix in the models shown in (E-H) with additional structure predictions obtained with PSIPRED 4.0 and AlphaFold2. Residues predicted to be part of the helix are highlighted in dark red (model), red (high confidence), and orange (medium confidence). The part of the helix that aligns with the DNA dimer is marked by a grey box. The 3 main residues that make contact with the DNA are underlined. Helix predictions for SCRM and MUTE are consistent between the 3 methods and always short. AlphaFold predicts an extension of the DNA-binding helix for SPCH and FAMA, which could interfere with nucleosome binding. Interestingly, in the PSIPRED prediction, this extension of the FAMA helix is interrupted by a stretch with very low helix probability, which could manifest as a flexible loop that breaks the rigid helix, thereby allowing nucleosome binding. bHLH, basic helix–loop–helix; SCRM, SCREAM; SPCH, SPEECHLESS. (TIF)

**S1 Table. Label-free quantification values and spectral counts of chromatin remodelers and coactivators identified as putative partners of stomatal lineage bHLH dimers.** (XLSX)

**S2 Table. Primers used to create plasmids used in this work.** (DOCX)

**S1 Data. Compilation of source data underlying all figures.**
(XLSX)

## Author Contributions

**Conceptualization:** Ao Liu, Andrea Mair.

**Data curation:** Ao Liu.

**Formal analysis:** Ao Liu, Andrea Mair.

**Funding acquisition:** Dominique C. Bergmann.

**Investigation:** Ao Liu, Andrea Mair, Juliana L. Matos, Macy Vollbrecht, Shou-Ling Xu.

**Resources:** Shou-Ling Xu, Dominique C. Bergmann.

**Supervision:** Dominique C. Bergmann.

**Visualization:** Ao Liu, Andrea Mair.

**Writing – original draft:** Ao Liu, Andrea Mair, Dominique C. Bergmann.

**Writing – review & editing:** Ao Liu, Andrea Mair, Juliana L. Matos, Dominique C. Bergmann.

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
