## [Editor Report · Decision Letter 0]

8 Jan 2024

Dear Dr Bergmann, 

Thank you for submitting your manuscript entitled "Cell Fate Programming by Transcription Factors and Epigenetic Machineryin Stomatal Development" for consideration as a Research Article by PLOS Biology.

Your manuscript has now been evaluated by the PLOS Biology editorial staff as well as by an academic editor with relevant expertise and I am writing to let you know that we would like to send your submission out for external peer review.

Once your full submission is complete, your paper will undergo a series of checks in preparation for peer review. After your manuscript has passed the checks it will be sent out for review. To provide the metadata for your submission, please Login to Editorial Manager (https://www.editorialmanager.com/pbiology) within two working days, i.e. by Jan 10 2024 11:59PM.

Kind regards,

Ines

--

Ines Alvarez-Garcia, PhD

Senior Editor

PLOS Biology

---

## [Decision Letter · Decision Letter 1]

18 Mar 2024

Dear Dr Bergmann,

Thank you for your patience while your manuscript entitled "Cell Fate Programming by Transcription Factors and Epigenetic Machinery in Stomatal Development" was peer-reviewed at PLOS Biology. Please also accept my apologies for the time it has taken us to provide you with a decision. The manuscript has now been evaluated by the PLOS Biology editors, an Academic Editor with relevant expertise, and by two independent reviewers. 

The reviews are attached below. As you will see, the reviewers find the conclusions novel and interesting for the field, but they also raise several issues that would need to be addressed before we consider the manuscript for publication. Reviewer 1 mostly suggests minor improvements in the figurers and asks for several clarifications, whereas Reviewer 2 is more critical and suggests an experiment expressing FAMA to see if it has a similar effect to MUTE. While this reviewer could have misinterpreted the arguments considered, we think this experiment would be important to clarify whether or not the differences in binding profile between MUTE and FAMA translate into differences in the ability to displace nucleosomes. In addition, the Academic Editor thinks that the section describing the differences between FAMA and MUTE is hard to interpret, thus we would encourage you to improve the writing to make the argument clearer to readers. The Academic Editor is also concerned that the differences between FAMA and both MUTE and SCRM might be technical because SCRM and MUTE profiles come from previous datasets, whilst FAMA profile is from a database generated for this manuscript. Thus, we think you should present in the main results section some data to confirm that the new profiling data is comparable – in terms of coverage, signal to noise, background signal, number of peaks, etc. In an ideal situation, the MUTE or SCRM profiling could be repeated to demonstrate that technical variability does not account for the differences between their profiles and that of FAMA.

In light of the reviews, which you will find at the end of this email, we would like to invite you to revise the work to thoroughly address the reviewers' reports and Academic Editor's comments. Given the revision needed, we cannot make a decision about publication until we have seen the revised manuscript and your response to the reviewers' comments. Your revised manuscript is likely to be sent for further evaluation by all or a subset of the reviewers.

**IMPORTANT - SUBMITTING YOUR REVISION**

3. Resubmission Checklist

a) *PLOS Data Policy*

b) *Published Peer Review*

Sincerely,

Ines

--

Ines Alvarez-Garcia, PhD

Senior Editor

PLOS Biology

Reviewers' comments

Rev. 1:

The manuscript by Liu et al, "Cell Fate Programming by Transcription Factors and Epigenetic Machinery in Stomatal Development", investigated how MUTE, SCRM and PAMA, key bHLH TFs in stomatal development, interplay with chromatin at their target genes and how they functionally interplay with other epigenetic factors to control gene expression.

The authors first analyzed transcriptional entropy in stomatal lineage single cell RNA-seq data, and identified different phases of transcriptional regulation during stomatal development, which they intended to use as indicators of differentiated vs undifferentiated cells . They further showed that MUTE, FAMA and SCRM could indeed overcome repressive chromatin and each has the potential to induce nucleosome repositioning during these phases. Lastly, the team used TurboID-based strategy to identify the proteins that are physically associated with FAMA by LC-MS and found BRM complex components and HAC1, a histone acetyl transferase. They then demonstrated that stomatal lineage-specific knock down of components of the BRM complex or HAC1 results in reduced expression of the bHLHs' targets and plants that are impaired in forming terminally differentiated stomata. It is a work of significant interest to the plant biology and epigenetics community; and in my view, the major contribution of this work is the identification of BRM and HAC1 and demonstration of their physical association with the bHLHs and their functional interplay at target genes in controlling stomata development. Overall, the manuscript is well written and the evidence presented support their conclusion.

I only have a few minor suggestions for improving the manuscript:

Figure 1a: what is the X-axis? Please label or explain.

Lines 147-156 and Figure 2 and Figure S3: the authors analyzed and compared previously published

ChIP-seq data with their newly generated ChIP-seq data, as well as scRNA-seq data. Please add some background information indicating that these experiments were conducted with similar plant materials including growth conditions so they are comparable and they are suitable for this manuscript. This also applies to other places in this manuscript where published data were used.

Table 1: about the TurboID-based MS identification of SCRM interactors: should the numbers of peptides identified by MS for each of the interactors listed? Such data would not only show the interaction, but can also show it in a quantitative manner.

Lines 241-244: the authors said "Interaction between the bHLHs and HAC1 was relatively weak in both Y2H (Fig S5a) and bimolecular fluorescence complementation (BiFC) assays (Fig S5c), suggesting that strong interaction with HAC1 may require the presence of the heterodimer or an additional linker protein (e.g. LUH, which interacts both with SCRM and FAMA and with HAC1 (Fig S5a and b))." This statement is true for Y2H, but not for BiFC as it is done in planta.

Figure 5 and the related text: about the BRM and HAC1 amiRNAi lines, the authors did not show in this manuscript to what extent the BRM/HAC1 transcript or protein were reduced? It would be helpful to present such data. It seems that they were using their previously generated lines, but it would be helpful to confirm again, as it might change over time. However, I did notice that in the Materials and Methods section, they described the RNA extraction and qRT-PCR analysis of these lines, but I did not see any mentioning of such results in either the figures or the main text.

Line 298: cite references for "…a line we termed 'short-lived FAMA…',".

Rev. 2:

The ms by Liu et al. on cell fate reprogramming by TFs and their recruitment of epigenetic machinery in stomata provides thoughtful experiments on a key aspect of plant gene regulation—the identification of pioneering transcription factors that can access closed chromatin. This is a key mechanism in development that determines how cells embark on new fates. The paper contains a series of key experiments that provide evidence for what may be perhaps the second good case for pioneering transcription factors in plants. The experiments themselves are rigorous and appropriate tests for this major claim. The argument is strong, and, overall, the chain of evidence, the logic of the experiments, and the presentation and interpretation of the results are all excellent. I have one major question on the chain of evidence that I cannot reconcile and weakens the argument if my interpretation is correct. It may not be, in which case, I feel the ms needs some clarification. I also have several minor and easily addressed comments.

1. In the section on bHLHs having the potential to bind closed chromatin and remodel it (starting 178), the evidence showed that FAMA (but not SCRM or MUTE) had comparable affinity to nucleosome and non-nucleosome bound chromatin. This, together with binding co-occurrence data led to the conclusion that FAMA and SCRM together bind nucleosomal DNA. But then, in experiments that built on the evidence to show the potential to remodel, the authors focused on MUTE. The inducible MUTE experiment followed by MNase nucleosome occupancy analysis supported its role to change nucleosome density. To me, this was a disconnect. The binding affinity pointed to FAMA as the pioneer factor with SCRM implicated as a partner. But then the functional test was done on MUTE, for which affinity evidence did not implicate? So, this breaks the chain of evidence in my mind that build the case for FAMA as a pioneering factor. The remodeling test would need to be done on FAMA, not MUTE? Could the authors address this issue?

Minor Points

2. I struggled to try to understand how the entropy was measured until I carefully read the methods. But it's such a lengthy part of the set up that I think it should be clear in the text. E.g., I wondered whether entropy was being measured from perspective of a single gene among all the cells or among all the genes. The seems to be the case but clear explanation in the main text would help.

3. Fig. 2D, it looks like SPCH should be significant in the opposite direction from MUTE, FAMA, and SCRM. I couldn't see how these statistics were done, one tailed?

4. I got a bit hung up on the FAMA_LGK experiment. It's not clear what is progression of events is assumed to be happening in the reprogramming. Are cells never reaching the mature state or do they get there and slip back, such that remodeling would have occurred and then reverted?

---

## [Decision Letter · Decision Letter 2]

13 Jun 2024

Dear Dr Bergmann,

Thank you for your patience while we considered your revised manuscript entitled "Cell Fate Programming by Transcription Factors and Epigenetic Machinery in Stomatal Development" for publication as a Research Article at PLOS Biology. This revised version of your manuscript has been evaluated by the PLOS Biology editors, the Academic Editor and one of the original reviewers.

Based on the reviews, we are likely to accept this manuscript for publication, provided you satisfactorily address the data and other policy-related requests stated below.

In addition, we would like you to consider a suggestion to improve the title:

"bHLH transcription factors cooperate with chromatin remodelers to regulate cell fate decisions during Arabidopsis stomatal development"

We expect to receive your revised manuscript within two weeks. 

*Published Peer Review History*

*Press*

Sincerely,

Ines

--

Ines Alvarez-Garcia, PhD

Senior Editor

PLOS Biology

Fig. 1A, C, D; Fig. 2A, D; Fig. 3A, C, D; Fig. 4A, D, E; Fig. 5A; Fig. S1B; Fig. S3A, B; Fig. S4A-D; Fig. S6A, C, D and Fig. S8B

CODE POLICY

Reviewers' comments

Rev. 2: Ken Birnbaum - note that this reviewer has signed his review

The authors have addressed my comment. The revisions to the nucleosome vs. non-nucleosome binding section clarified my misinterpretation of the result.

---

## [Editor Report · Decision Letter 3]

26 Jul 2024

Dear Dr Bergmann,

Thank you for the submission of your revised Research Article entitled "bHLH transcription factors cooperate with chromatin remodelers to regulate cell fate decisions during Arabidopsis stomatal development" for publication in PLOS Biology. On behalf of my colleagues and the Academic Editor, Peter Sarkies, I am delighted to let you know that we can in principle accept your manuscript for publication, provided you address any remaining formatting and reporting issues. These will be detailed in an email you should receive within 2-3 business days from our colleagues in the journal operations team; no action is required from you until then. Please note that we will not be able to formally accept your manuscript and schedule it for publication until you have completed any requested changes.

PRESS

Sincerely, 

Ines

--

Ines Alvarez-Garcia, PhD

Senior Editor

PLOS Biology
